# Experimental and Numerical Simulation of the Formation of Cold Seep Carbonates in Marine Sediments

**DOI:** 10.3390/ijerph16081433

**Published:** 2019-04-22

**Authors:** Tao Ye, Guangrong Jin, Daidai Wu, Lihua Liu

**Affiliations:** 1Guangzhou Institute of Energy Conversion, Chinese Academy of Sciences, Guangzhou 510640, China; yetaook@gmail.com (T.Y.); jingr@ms.giec.ac.cn (G.J.); wudd@ms.giec.ac.cn (D.W.); 2CAS Key Laboratory of Gas Hydrate, Guangzhou 510640, China; 3Guangdong Provincial Key Laboratory of New and Renewable Energy Research and Development, Guangzhou 510640, China; 4Guangzhou Center for Gas Hydrate Research, Chinese Academy of Sciences, Guangzhou 510640, China; 5University of Chinese Academy of Sciences, Beijing 100049, China; 6Institution of South China Sea Ecology and Environmental Engineering, Chinese Academy of Sciences, Guangzhou 510640, China

**Keywords:** cold seeps, authigenic carbonates, experimental and numerical simulation, diagenesis rate, authigenic carbonate, precipitation rate

## Abstract

Cold seep emissions of low temperature fluid from the marine sediment basins are mainly comprised of methane and other hydrocarbons. A series of biogeochemical processes related to methane lead to the formation of authigenic carbonate minerals. In this study, a self-built experimental device was used to study the formation process of carbonate minerals under cold seep conditions. The concentrations of pore water chemicals, HCO_3−_ and Ca^2+^ at different heights of the reactor under flow conditions can be observed. According to the experimental results, the formation process of carbonate minerals under cold seep conditions was estimated, that 1 m carbonate growth needs 12,000 and 7000 years, respectively, under fast (5 mL·min^−1^) and slow emission (1 mL·min^−1^) conditions. Furthermore, TOUGHREACT was used to simulate the diagenesis process. A 1D unsteady react-transport model was developed, and the experimental data was used to constrain the simulation. The results of simulation show that the carbonates need 17,000 and 9700 years to grow 1 m under the condition of fast and slow flow scenarios, respectively. The results of this work will contribute to the study of foundation on the formation of authigenic minerals in cold seep areas, and for the physical properties of sedimentary media as well.

## 1. Introduction

Cold seep is a phenomenon involving emissions of methane-rich fluid, occasionally accompanied by heavier hydrocarbons and CO_2_, from marine sediments. The temperature of the fluid is close to the temperature of the surrounding seawater [1]. Most of the ascending methane is consumed by anaerobic oxidation (AOM) mediated by anaerobic methanotrophic archaea (ANME) during the fluid migration in marine sediments. Unconsumed methane may enter the water column or even reach the atmosphere, causing an increase in the concentration of methane in the atmosphere, which due to its strong greenhouse effect may affect regional and global climates [2].

The AOM process produces the bicarbonate and changes the pore water composition and pH, then bicarbonate react with metal ions such as calcium and magnesium and eventually forms aragonite, high-magnesium calcite and less commonly low-magnesium calcite and dolomite [3,4]. Cold seep carbonate formation involving microbial activity is which very complicated, although the following simplified reaction equation can be proposed: (1)$${\mathrm{Ca}}^{2+}{+2\mathrm{HCO}}_{3-}↔{\mathrm{CaCO}}_{3}\left(s\right){+\mathrm{CO}}_{2}{+H}_{2}O$$

The above precipitation and related biogeochemical processes alter the composition of minerals and compounds in sediments. Therefore, the cold seep carbonates preserve the characteristic information of the composition of cold seep fluids and the flux of fluids during the formation of minerals [5,6]. The formation of cold seep carbonates also changes the porosity, density and permeability of the medium, which in turn affects fluid migration and the stability of the sediments [2,7,8]. Cold seep carbonates are one of important carbon sinks that bury a large amount of carbon [9], which is also significant for the study of the carbon cycle [10] and regional marine ecology events such as acidification and other issues [11].

Unfortunately, we still know little about the kinetics of cold seep carbonate precipitation, especially related to the diagenesis rate, even in the field, although there have been some achievements. Godinho and Williams imaged the evolution of calcite precipitation using a flow- through system and the time-lapse X-ray computed tomography, which indicated that reactive crystal surfaces within less permeable regions grow at a slower rate than that expected from the bulk fluid composition, but they did not consider the situation under natural conditions [12]. Bracco et al. modeled the macroscopic rates under a range of alkaline solution conditions measured by atomic force microscopy (AFM), however, this approach still cannot model equilibrium or pH effects [13]. In addition, Zuddas et al. characterized the effects of ionic strength on the kinetics of calcite precipitation from seawater [14].

As mentioned above, the formation of cold seep carbonates, which is a complex biogeochemical process involving microorganisms, ion interactions and changes in porous media, occurs in the natural high-pressure and low-temperature marine environment. Moreover, in the natural cold seep areas, the situation would be more complicated than this due to fluid flow and bioturbation [4,15,16]. Therefore, as in situ observations are costly and have not been reported, experimental and numerical simulation represent an important research method.

Previous simulations have been carried out, for instance, Krause et al. [17] found that microbial extracellular polymers participate in the precipitation of Mg-rich dolomite under modern seawater conditions. Liu [18] observed the formation of cold seep authigenic minerals under steady-state conditions, and verified the relationship between AOM process and cold seep carbonate formation. However, the dynamic mechanism of minerals under the flow conditions of cold seeps still needs to be further explored. Luff et al. calculated the rate of cold seep carbonate simulation and the effect of carbonate crusts on fluids, but there are no relevant experiments to support it [19]. The flow-through experiment conducted by Steeb et al. [20] provided important kinetic information about biogeochemical reactions in natural sediments, but their study lacked in-depth precipitation of subsequent cold seep carbonates. This work intends to simulate the precipitation process of cold seep carbonates in marine sediments under cold seep flow conditions, and use numerical simulation tools to model the experimental data to further explore the kinetic process of cold seep carbonates formation.

## 2. Materials and Methods 

### 2.1. Experimental Simulation of Fluid Leakage Reaction

#### 2.1.1. Experimental Device

A self-built biogeochemical simulation experimental device has been set up to model the formation of carbonates (Figure 1), which consists of a liquid supply system, reactor, and sample collection subsystems. Specifically, the liquid supply subsystem pumps the solution from the liquid storage tank into the reaction subsystem. The reaction subsystem consists of a plexiglass reactor with an inner diameter of 80 mm, an outer diameter of 110 mm, a height of 300 mm and an effective volume of 1500 mL, and the reactor is placed inside an incubator to maintain a constant pressure and temperature (0–200 °C, 0–20 MPa). The sample collection subsystem consists a line of sampling ports on the side of the reactor for sampling at different heights.

#### 2.1.2. Experimental Conditions

The experimental system was set according to the field data of cold seeps in the northern slope of the South China Sea. Specifically, we refer to the pore water data of the 0–100 cmbsf sediment pore water sample (D17-2015) at the depth of the natural gas hydrate development area of Dongsha [6] (Table 1). The relevant parameters are shown in Table 1.

The experimental simulation started with a simple condition and laid the foundation for further work under complex conditions. Quartz sands were used as deposition media to simulate the sea floor sediments. The Ca^2+^ concentration (50 mM·L^−1^) was increased to replace the interaction of Mg^2+^ and Ca^2+^ in seawater (table 1). The alkalinity of the solution in this experiment was all derived from HCO_3−_, and the concentration of HCO_3−_ was determined to be 3 mM·L^−1^ with reference to Table 2. The pH value is set to 8 referring to the cold seeps area [18]. The experiment was currently run under atmosphere pressure (0.1 MPa). Equation (2) was used to calculate the pressure changes in this system:(2)P=F/S
where P is the pressure, F is the force (simplified to gravity in this system), and S is the bottom area of the reactor. The pressure difference between the top and bottom interfaces is less than 0.004 MPa. We thus ignore the impact of pressure changes.

The percolation velocity of cold seeps fluids at the bacteria mat can reach up to 10 m·a^−1^ [21], while the maximum value is over 1000 m·a^−1^ when it erupts [22]. In this paper, we adopted the middle values of 5 mL·min^−1^ and 1 mL min^−1^ (equivalent to 522 m·a^−1^ and 104 m·a^−1^) to inject the HCO_3−_ solution into the reactor. The formation kinetics of cold seep carbonates are thus simulated. The ionic strength of the pore water at different flow rates, as well as the diffusion, advection and the contact time of the two reactants are also different. In this study, the calcite saturation index was calculated to be 0.41 using PHREEQC, indicating that calcite precipitation was happening during the experiment.

#### 2.1.3. Experimental Steps

The experimental processes are briefly described as follows:

Analytically pure anhydrous calcium chloride and sodium bicarbonate powder were used to prepare the solution prior to the experiment. The solution was quickly stored in volumetric flasks to avoid possible reaction with substances in the air. The system isolated from air during the subsequent experiments. The quartz sand is soaked in dilute hydrochloric acid for 24 hours in order to remove impurities, repeatedly washed by pure water, followed by drying at 45 °C for 24 hours to ensure that the hydrochloric acid has been removed. Secondly, the quartz sand was placed in the reactor and compacted (the quartz sand column height was about 30 cm). The porosity of the quartz sand in the reactor was 0.42 by the immersion method, which was close to the field data of 0.424 [23]. The reactor was placed in an incubator and the temperature was 15 °C. The 50 mM·L^−1^ CaCl_2_ solution was injected into the reactor after the quartz sand was filled by siphoning to ensure that the gas in the sand column was sufficiently removed. After the sand column is saturated and the temperature is stable, a 3 mM·L^−1^ NaHCO_3_ solution (adjusted to pH = 8) is pumped from the bottom of the reactor at a constant rate (Table 3) using a constant speed constant pressure pump to simulate the upward migration of the HCO_3−_, the fluid which is same as the produce of AOM in the pore water. After the solution began to be injected, the change in ionic composition in the solution was measured by a lateral sampling port interval of 20 min (Figure 2), approximately 7 mL per sample. Each experiment run 160 min.

#### 2.1.4. Analytical Method

After the experiment, the quartz sand was dried at 45 °C for 24 hours after washing with pure water. The white crusted material on the surface of the sand was carefully collected and then subjected to an X-ray diffraction (XRD) analysis at the Guangzhou Institute of Energy Conversion (GIEC) of CAS. The concentration of HCO_3−_ was determined by acid-base neutralization titration in units of mg·L^−1^. The Ca^2+^ was carried out at GIEC, using an Inductive Coupled Plasma Emission Spectrometer (ICP, Model OPTIMA 8000DV, PERKINELMER, Waltham, Massachusetts, USA) in mg·L^−1^.

### 2.2. Numerical Simulation Method

The simulation was performed by TOUGHREACT, which is a numerical simulation tool for non-isothermal flow chemical reactions of multiphase fluids in pores and fractured media [24]. Some studies have used TOUGHREACT to investigate AOM and carbonate precipitation in cold seep areas [18,25,26]. A brief introduction of the numerical simulation process is provided below, and for details, readers should see the TOUGHREACT Handbook [24,27].

#### 2.2.1. Set Up of the Constitutive Equation

The mass balance equations established for the experimental carbonate precipitation in this paper are as follows:(3)$$\frac{\partial M_{k}}{\partial t}=-\nabla F_{K}+q_{k},$$
where *M* is the mass or energy accumulation, the unit is kg m^−2^, *F* is the mass flux, the unit is kg m^−2^ s^−1^, and *q* is the source and sink term in kg m^−2^ s^−1^.

#### 2.2.2. Transport and Diffusion of Solutes

The change in solution concentration is mainly controlled by diffusion, advection and reaction. The advection is represented by the following formula in the mass transfer process:(4)$$m_{adv}^{j}=\frac{\Delta t}{\mathrm{Vn}}\cdot \sum_{m}A_{nm}{\cdot u}_{nm}\cdot C_{nm}^{j}$$
where $m_{adv}^{j}$ is the concentration at which the main ions change due to convection, Δt is the time step, Vn is the volume of the nth grid, $A_{nm}$ represents the contact area of two adjacent grids *n* and *m*, and $u_{nm}$ is the Darcy velocity, $C_{nm}^{j}$ is the concentration of the jth ion at the n- and m-mesh contact faces.The formula for transferring matter by diffusion is:(5)$$m_{diff}^{j}=\frac{\Delta t}{\mathrm{Vn}}\cdot \sum_{m}A_{nm}\cdot D_{nm}\cdot \left[\frac{C_{m}^{j}-C_{n}^{j}}{d_{nm}}\right]$$
where $m_{diff}^{j}$ is the amount of the *j^et^* main ion transported by diffusion, $D_{nm}$ represents the diffusion coefficient in the water phase, Δt is the time step; *V_n_* is the volume of the *n^th^* grid, and $d_{nm}$ represents two adjacent. The distance between the grid *n* and *m*, *u_nm_* is the Darcy velocity, $\left[\frac{C_{m}^{j}-C_{n}^{j}}{d_{nm}}\right]$ represents the concentration of the *j^th^* ion at the *n* and *m* mesh contact faces.

#### 2.2.3. Chemical Reactions and Mineral Precipitation

For the mineral dissolution and precipitation, the kinetic reaction rate is expressed based on the transition-state theory of [28,29]:(6)$$r_{n}=\pm k_{n}A_{n}{\left|1-\left(\frac{Q}{k_{n}}\right)\right|}^{\eta }$$
where η is a positive value indicates mineral precipitation, a negative value indicates mineral dissolution, n is the reaction mineral number, *k_n_* is the reaction rate constant affected by temperature, *an* is the reaction specific surface area of mineral n, cm^2^·kg^−1^, *K_n_* is the equilibrium constant of mineral n under the specified temperature and pressure conditions, and *Q* is the ion activity. The product of the degree, *e* and *n*, is the experimental regression constant and is usually set to 1.0.

### 2.3. Initial and Boundary Conditions

Physical initial conditions: The mode includes solid and liquid phases, given temperature and pressure parameters according to experimental data. The interface between seawater and sediment is the upper boundary, and the lower boundary simulates the sulfate-methane conversion zone (SMTZ). The initial conditions and upper and lower boundary conditions are shown in Table 3.

The experiment was carried out under normal pressure, and the simulated sediment column length was only 30 cm, so that the difference of temperature and pressure at the top and bottom of the sediment column could be ignored, that is pressure was 0.1 MPa and the temperature was 15 °C under both upper boundary and lower boundary conditions. 

### 2.4. Meshing

In this paper, a 1D model was set referring to the actual dimensions of the reactor (height 30 cm, inner diameter 8 cm). The column is equally divided into 60 grids, each with a grid thickness of 5 mm.

## 3. Results and Discussion

### 3.1. Mineral Phase

The X-ray diffraction analysis was used to determine the mineral of the crust. Analysis conditions is as follow: starting position at 5.005 [°2Th.], end position at 69.989 [°2Th.], step size 0.008°, anode material is Cu:, voltage 40 kV, current 40 mA. After that, compare the diffraction pattern in the PANnalytical Example Database with HighScore Plus 3.0 software, made by PANalytical B.V., Netherlands as shown in Figure 3, where the black solid line indicates the experimental product and the solid gray line represents the calcite. 

The experimental precipitate highly matched calcite, with no redundant peaks appearing. Thus, CaCO_3_ in the experimental products is a calcite phase, and aragonite and dolomite are not detected. This result is consistent also with previous studies. In the solution, the aragonite is very unstable and can be spontaneously converted into calcite [30].

### 3.2. Experimental Results

The results of the rapid scenario (5 mL·min^−1^) are shown in Figure 2, the slow flow scenario (1 mL·min^−1^) are shown in Figure 2. In both Figure 2 and Figure 3, the *x*-axis and *y*-axis are Ca^2+^ and HCO_3−_ concentrations, respectively, in mM·L^−1^, and the *y*-axis is the height of the reactor in cm.

As Figure 2 shows, a sharp decrease in Ca_2+_ concentration and an increase in HCO_3_^−^ concentration at the same height (Figure 2a,b) when the HCO_3_^−^ solution was injected. This change implies carbonate precipitation. Figure 2 demonstrates the exhaustion of calcium ions at the bottom of the reactor, and the reaction zone gradually moves upward with time. The reaction zone was initially located at 0–5 cm (Figure 2a,b), then moved up to 2.5–7.5 cm at 80 min (Figure 2c,d), and later to 5–12.5 cm at 160 min (Figure 2e,f). It was observed that the moving speed of the reaction zone was relatively uniform and the interval of the reaction zone was gradually enlarged.

The results of the slow flow scenario (1 mL·min^−1^) are shown in Figure 2. The initial concentrations are the same as the fast flow scenario. The rapid decrease of Ca^2+^ concentration and the increase of HCO_3_^−^ concentration at the bottom of the reactor were observed at the first time period (Figure 2a,b). The reaction zone moved slightly upward with the bottom calcium ions depleted. The reaction zone was located at 0–2.5 cm at 20 min (Figure 2a,b), and rises to 0–5 cm at 80 min (Figure 2c,d), then to 2.5–7.5 cm at 140 min (Figure 2e,f).

### 3.3. Carbonate Precipitation Rate

The carbonate mineral precipitation rate ($R_{ppt}$) was calculated using Equation (7) based on the experimental results of [31,32]. We calculate the rate of calcite precipitation since no aragonite and dolomite has been determined during the experiment (see Section 3.1):(7)$$R_{ppt}=k_{ppt}\cdot {\left(\frac{\left[Ca^{2+}\right]\cdot \left[CO_{3}{}^{2-}\right]}{K_{sp}}-1\right)}^{n}$$
where $k_{ppt}$ is the precipitation kinetic constant of calcite, *k_PPT_* = 9.64 × 10^−3^ mM·cm^−2^ a^−1^; *K_sp_* is the thermodynamic equilibrium constant defining the solubility of aragonite or calcite. The solubility product constant of calcium carbonate at 15 °C is *Ksp* = 9.9 × 10^–6^ mM^2^·L^−2^ [33]; [Ca^2+^] is the dissolved calcium ion concentration, and [CO_3_^2−^] is the dissolved carbonate concentration. In a similar study of the predecessors (*n*), the value was between 0.4 and 2.8 [34]. The value of (*n*) adopted here was 1, the same as Luff et al. [32].

The dissolved carbonate concentration [CO_3_^2−^] in the solution was 2.34 × 10^−5^ mM·L^−1^ calculated by the PHREEQC version 3 based on database “PHREEQC.DAT” which is a computer program is designed to perform a wide variety of aqueous geochemical calculations. For the fast scenario (5 mL·min^−1^) the calcium carbonate precipitation rate at the bottom of the reactor (0 cm) is 204 μM cm^−2^ a^−1^ after 20 min of the solution injection according to Equation (7). That is, it takes about 12,000 years for the carbonates to grow 1 m under the fast flow scenario.

Similar to the fast flow scenario, the [CO_3_^2−^] in the slow flow scenario solution was 1.09 × 10^−5^ mM·L^−1^ after 20 min of HCO_3−_ solution injection (Figure 4). The precipitation rate of the carbonates at bottom (0 cm) of reactor (1 mL·min^−1^) was 354 μM cm^−2^ a^−1^. That is, it needs about 7000 years for the carbonates to grow 1 m under the slow flow scenario, which is faster than that of the fast flow scenario. This probably due to that the calcium ions at the bottom of the reactor (0 cm) are consumed rapidly during the first sampling (20 min) for the faster scenario, while the calcium ions in other areas cannot be replenished in time, resulting in lower reaction rate than the slow flow scenario. The precipitation rate of carbonate in the slow flow scenario shown in Figure 4 has four stages. Firstly, from 20–40 min, the precipitation rate reaches the maximum and remains basically the same (solid line), because the maximum supplement of the solutes. At 40–80 min, the formation rate reduced (dashed line). Then undergo a rapid decrease (dashed line) between 80–120 min. Finally, Ca^2+^ was depleted at 120–160 min, and the reaction is basically stagnant (open line). The limiting factor of the reaction, similar to the cold seep areas, should be the supply of Ca^2+^ and HCO_3−_ in the marine sediment since it is mainly replenished from sea water and by AOM.

### 3.4. Numerical Simulation of Carbonate Precipitation Rate

The concentrations of numerical simulation were shown in Figure 2 as indicated above marked by gray dots. The numerical simulation and experimental data of Ca^2+^ have the same trend and amplitude of concentration change as HCO_3−_.The area of calcium carbonate precipitation is basically the same as that of the experimentally observed reaction zone (Figure 2), while the experimental data in the reaction zone (dashed area) are faster than in the numerical simulation. The numerical simulation of carbonate precipitation rate was calculated using the same method as Equation (7). At 20 min, the fast flow scenario (5 mL·min^−1^) had a precipitation rate of 152 μM cm^−2^ a^−1^ at the bottom of the reactor (0 cm), or 17,000 years is needed for 1 m carbonates to form. In contrast, the slow flow scenario (1 mL·min^−1^) had a precipitation rate of 267 μM cm ^−2^ a^−1^ at the same depth, that is, the thickness of the carbonate rock increased by 1 m after around 9700 year. The same trend as shown in Figure 2. The simulation results of carbonate precipitation rate is also slightly slower than the experimental one (the fast flow scenario and the slow flow scenario are 204 μM cm^−2^ a^−1^ and 354 μM cm^−2^ a^−1^, respectively, and after about 7000 years and 12,000, carbonate grew by 1 m).

Our results are consistent with Raiswell’s for the formation of carbonates in marine sediments between 15 and 150 mbsf obtained by a diagenetic model [35]. The precipitation rate was 0.001–0.4 cm a^−1^, and 250–10,000 years are needed to grow 1 m. Ths is slightly faster than the precipitation rate of Luff, which was 160 μM cm^−2^ a^−1^. [32]. This difference may be caused by a faster flow rate and the sampling procedures.

Karaca et al. calculated the precipitation rate of cold seep carbonates in the cold seep area of Costa Rica by numerical simulation. Their results suggested that the production of carbonate alkalinity and the formation of authigenic carbonate in the middle flow rate (3 °C, 40 cm a^−1^) is the strongest, with reduced low and high flux conditions (0.1 and 200 cm a^−1^) [36]. The calculated precipitation rate of cold seep carbonate rocks is consistent with the continental margin near Costa Rica (40–310 μM cm^−2^ a^−1^) [37]. This is also similar to the results of this study, but slightly slower, which may be due to the faster flow rate used in this study, which will be discussed later.

As discussed above, several speculative reasons could contribute to the differences between the experimental and numerical results. First, sampling may distribute the flow path and speed up the fluid surge during the experiment, which led to an accelerated reaction process. Secondly, fluids may tend to flow along the walls of the reactor during the experiment [38], which also accelerates the flow rate of the fluid, resulting in a faster reaction rate in the experiment. Moreover, during the experiment, solutes diffuse simultaneously in the vertical and radial directions. The 1D model considers only the vertical diffusion, so that the reaction rate of the experimental observation maybe faster. Finally, calcium ions and bicarbonate were diluted due to the increase in the total volume of the solution, which also led to experimentally observed reaction rates being faster than ideal conditions.

## 4. Conclusions

A self-developed biogeochemical experimental device was used to carry out an experiment to simulate the diagenesis of cold seep authigenic carbonates under different flow scenarios. The bicarbonate solution was injected into the reactor containing calcium ion solution at a constant rate of 5 mL·min^−1^ and 1 mL·min^−1^, respectively, followed by sampling the pore water at different reactor heights. 

This study starts with relatively simple conditions, such as under normal pressure and without involvement of microorganisms. However, the experimental data under such conditions could shed some light on the carbonate sedimentation patterns under different flow rates. It can also provide a comparison for experiments with microbial participation in the future to explore the role of microorganisms in the experimental process.

The formation rate of cold seep carbonate was estimated based on the experimental results. At 20 min after the start of the injection process, the precipitation rates of carbonates were 204 μM cm^−2^ a^−1^ and 354 μM cm^−2^ a^−1^ under the two flow scenarios. The precipitation rate calculated in this paper is consistent with the results of Raiswell et al. and Karaca et al. [35,37], and slightly higher than the simulation result of Luff et al. 120 μM cm^−2^ a^−1^ [32].

The TOUGHREACT software was used to simulate the mineralization process of cold seep carbonates under the cold seep conditions constrained by the experimental results. The simulated deposition rate is slightly slower than the experimental results. Specifically, the growth rate of carbonates is 152 μM cm^−2^ a^−1^ (5 mL·min^−1^) while 267 μM cm^−2^ a^−1^ (1 mL·min^−1^). The different probably due to the sampling distribution and shortcuts of the fluid flow. The combination of experimental and numerical simulation adopted in this study could reveal the formation of authigenic minerals in natural environments under similar conditions, especially for long time scales.

The calculated precipitation rate did not consider the crystallization of minerals and the mineral specific surface area on mineral formation [39], nor the activity of microorganisms, meaning that the actual precipitation rate may be slower than the laboratory measurements. The complex composition of porewater, magnesium ions, phosphates, pH and temperature stress, affect the in-situ carbonates deposits as well [39,40]. Further work is needed for fill the gap of our knowledge.

The conditions used in the experimental and numerical simulations are simple compared with the field, since the fluid flow and bioturbation in the actual environment were ignored here. However, these factors have a great influence on the kinetic parameters of the early diagenesis of the cold seep carbonates. In the future, the bio-geochemical experiments with microbial participation in natural sediments can be used to observe the mineral precipitation process under the action of microorganisms.

## Figures and Tables

**Figure 1 ijerph-16-01433-f001:**
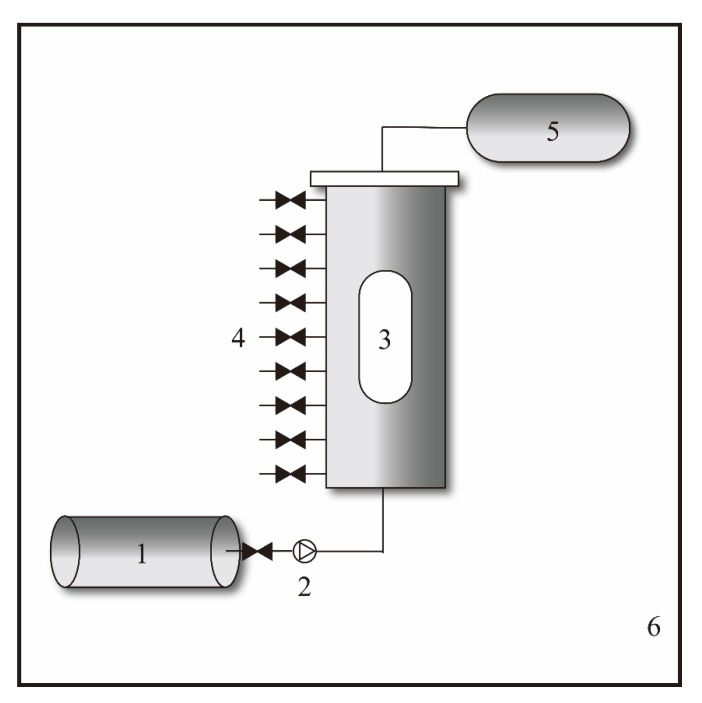
Biogeochemical simulation experiment device. where 1 is a liquid storage tank; 2 is a constant speed constant pressure pump; 3 is a sapphire window; 4 is a sampling port; 5 is a liquid discharge metering system; 6 is a constant pressure incubator.

**Figure 2 ijerph-16-01433-f002:**
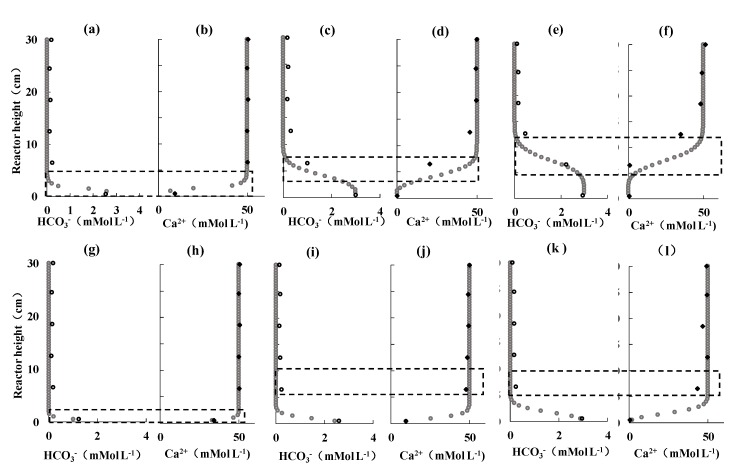
Fast scenario and slow scenario partly experiment and numerical simulation results. The first row of Figure 2 depicts the data of experiments and numerical simulations of the fast flow scenario, where (**a**), (**c**), and (**e**) are bicarbonate concentration at 20 min, 80 min and 160 min, after the solution injection respectively, fast flow scenario, similarly, the (**b**), (**d**), (**f**) are calcium ion concentrations; The second row is the data of slow scenario and the numerical simulation. Where (**g**), (**i**), (**k**), and (**h**), (**j**), (**l**) are the experimental and numerical simulation results of the bicarbonate and calcium ion concentrations in the slow flow scenario at 20 min, 80 min and 160 min at 20 min, 80 min and 160 min, after the solution injection respectively. The hollow dot is HCO_3−_ concentration and the solid square is Ca^2+^ concentration in the experiment while the gray dot is the data of simulation. The dashed area is the area we assume that the hypothesis reaction takes place in the reactor (reaction zone).

**Figure 3 ijerph-16-01433-f003:**
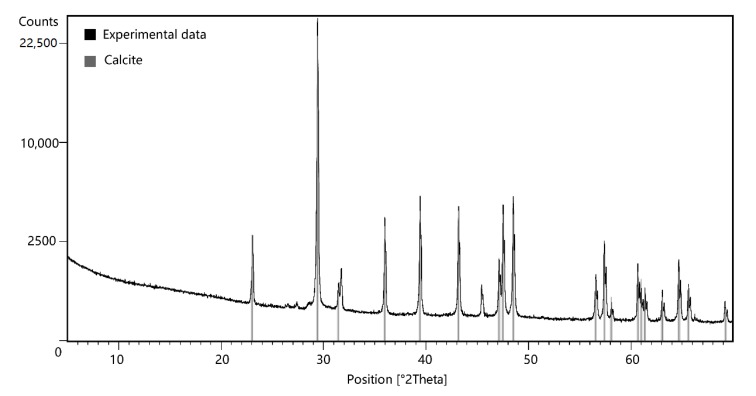
X-ray diffraction pattern of experimental product.

**Figure 4 ijerph-16-01433-f004:**
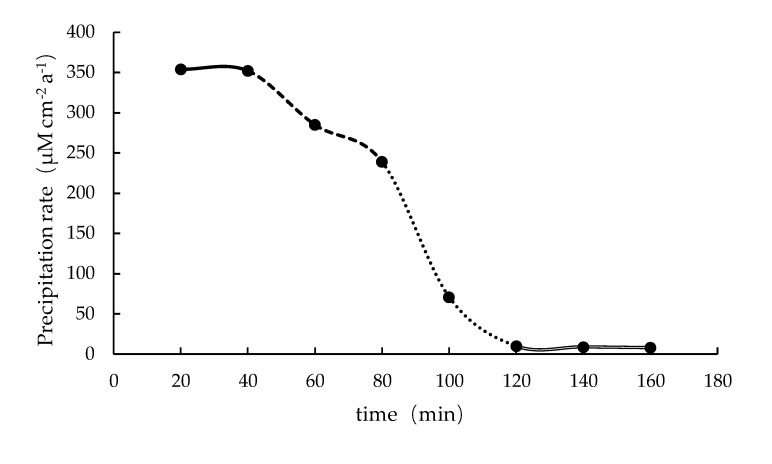
Carbonate precipitation rate at 0 cm of reactor in slow flow scenario.

**Table 1 ijerph-16-01433-t001:** Parameters adopted in the experiment.

	Ca^2+^ (mM)	Fe^2+^ (µM)	Na^+^ (mM)	Mg^2+^ (mM)	ALK (mM)	Cl-(mM)	SO_4_^2−^ (mM)	Salinity (PSU)	Temperature (°C)
**D17-2015**	10.16	0.45	475	53.8	3	556	2	35	15.6
**Parameters used in the experiment**	50	0	470	0	3	490	0	35	15

Data from [6].

**Table 2 ijerph-16-01433-t002:** Experimental condition.

Experimental Condition	Temperature (°C)	Pressure (MPa)	Solution in the Reactor	Infused Solution	Injection Rate (mL·min^−1^)	Sampling Interval (min)	Number of Samples
Fast flow scenario	15	0.1	50 mM CaCl_2_	3 mM NaHCO_3_	5	20	8
Slow flow scenario	15	0.1	50 mM CaCl_2_	3 mM NaHCO_3_	1	20	8

**Table 3 ijerph-16-01433-t003:** Initial conditions and upper and lower boundary conditions.

Initial Condition		
Solution Composition	Concentration (mM)	Injection Rate (mL·min^−1^)
CaCl_2_	50	-
NaHCO_3_	3	
**Upper boundary condition**		
CaCl_2_	50	0.6 or 2.7
NaHCO_3_	0	
**Lower boundary condition**		
CaCl_2_	0	-
NaHCO_3_	10	1 or 5

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
