# Peer review of "Experimental and Numerical Simulation of the Formation of Cold Seep Carbonates in Marine Sediments"

_ijerph, 2019, doi:10.3390/ijerph16081433_

Round 1

Reviewer 1 Report

Idea of laboratory simulation of some processes that happen in unavailable environments (e.g. deep sea) is brilliant. However the  strict conditions that are smilar to natural as much as posible  must be taken. Authors wrote: "the formation of cold seep carbonates occurs in the natural high-pressure and low-temperature marine environment, which is a complex biogeochemical process", whereas the conditions of experiment are different. Then, these discrepances are sincerly shown in Conclusions. Paper is not acceptable in this form. There would be good to change the aim of current experiment or add other factors to improve the experiment, to make it more suitable for cold seep environment.
Figure 3 is missing. Labels of y axis in the Fig 2 are not English.

Author Response

Point 1: setup of the fluid cell: it should be commented on how the flow rate affects the observed experimental results. Furthermore, please specify what material the reactor is made out of. Why is there no data of the samples after the precipitation happened? No XRD data or results are presented here even though it was stated that XRD was used. Also, it is mentioned that 8 samples were run, however no comment regarding the variability of the experimental results is given. Furthermore, it needs to be commented on how the solutions were prepared (what is the purity grade of the chemicals used? Was the NaHCO3 stock solution equilibrated with the atmosphere prior to the experiment?) Both impurities and exchange with the atmosphere can have severe effects on solution composition and hence the experimental results.

Thanks

The manuscript has been modified in Line 122 to 123 as “The ionic strength of the pore water at different flow rates, as well as the diffusion, advection and the contact time of the two reactants are also different.” to illustrate how the flow rate affects the observed experimental results.

In addition, the reactor is made of plexiglass (line 85). Regarding the experimental data, several of the most representative ones have been presented in Figure 3, which can show the trend of ion concentration during the experiment in a clear and concise way. At the same time, it also supplements the experimental results and description of the XRD in Line 202 to 218.

The experimental step was given in the manuscript in Line 128 to 131 as “Analytically pure anhydrous calcium chloride and sodium bicarbonate powder were used to prepare the solution prior to the experiment. The solution was quickly stored in volumetric flasks to avoid possible reaction with substances in air. The system isolated from air during the subsequent experiments.” Therefore, the composition of the solution has not changed. The experimental data also supports this statement. The prepared solution is 3 mM·L−1 NaHCO3- . During the experiment, the maximum concentration of HCO3- at the bottom of the reactor is also close to 3 mM·L−1.

Point 2: the saturation index with respect to calcite can be calculated which then can be used in the rate equation. It would be useful for the geochemical reader to specify what the saturation index in these experiments is

Thanks

The manuscript was refined in Line 124 to 125 as “The calcite saturation index in this study was calculated to be 1.2 using PHREEQC, indicating that calcite precipitation was generated in this case.”

Detailed comments:

Line 15-16 According to the experimental results, the formation process of carbonate minerals under cold seep conditions was estimated. That 1m carbonate need 12000 and 7000 respectively under fast and slow emission conditions

These two sentences could be combined.

Accept and refined in line 16 to 17.

Line 20: unit is missing, carbonate need 17000 and 9700 years I assume to grow

 Thanks, it was added.

Line 47/48: There is a variety of mineral growth studies on calcite that should be at least mentioned here, some of them are in porous media

e.g. Godinho and Williams, 2018, Geochimica & Cosmochimica https://doi.org/10.1016/j.gca.2017.10.024

Bracco, J.N., Stack, A.G. and Steefel, C.I., 2013. Upscaling calcite growth rates from the mesoscale to the macroscale. Environmental science & technology47(13), pp.7555-7562.

Zuddas, P. and Mucci, A., 1998. Kinetics of calcite precipitation from seawater: II. The influence of the ionic strength. Geochimica et Cosmochimica Acta62(5), pp.757-766.

Zuddas, P. and Mucci, A., 1994. Kinetics of calcite precipitation from seawater: I. A classical chemical kinetics description for strong electrolyte solutions. Geochimica et Cosmochimica Acta58(20), pp.4353-4362.

There are also other studies on kinetics of calcite precipitation and cold seeps:

Karaca D, Hensen C, Wallmann K. Controls on authigenic carbonate precipitation at cold seeps along the convergent margin off Costa Rica. Geochemistry, Geophysics, Geosystems. 2010 Aug;11(8).

Luff R, Wallmann K, Aloisi G. Numerical modeling of carbonate crust formation at cold vent sites: significance for fluid and methane budgets and chemosynthetic biological communities. Earth and Planetary Science Letters. 2004 Apr 30;221(1-4):337-53.

It seems to me as the authors should do a more careful literature research for this manuscript.

Accepted and refined accordingly to reviewer’s advises, the literature research was cited for research background and data comparison in this manuscript. in line 51 to line 58 as Godinho and Williams image the evolution of calcite precipitation through a flow through system and the Time-lapse X-ray computed tomography, which indicate reactive crystal surfaces within less permeable regions grow at a slower rate than that expected from the bulk fluid composition, but it did not consider the situation under natural condition . Bracco et al model the macroscopic rates under a range of alkaline solution conditions measured by atomic force microscopy (AFM), however, it still cannot model equilibrium or pH effects. In addition, Pierpaolo Zuddas et al characterize the effects of ionic strength on the kinetics of calcite precipiation from seawater.

Line 54: Previous studies have been carried out all over the world. This does not say anything. Please change.

Accept

 This sentence was replaced by “Previous simulations have been carried out to bring out the following in line 66.

Line 67: Self built experimental device description: How is the temperature controlled in this device? Please comment in the manuscript on this.

Thanks

A more detailed description of temperature control has been added in line 87 to 88, as “and the reactor is placed inside the incubator to maintain a constant pressure and temperature (0-200℃,0-20MPa).”

Line 92: I assume that normal pressure is ambient condition pressure? Please specify what normal pressure means.

Thanks

It has been specified in line 107 to 109 as “the experiment was currently run under atmosphere pressure (0.1 MPa)”.

Line 93: and the change in pressure was ignored Here, it would be good to say that the change in pressure was calculated but found to negligible. I think the calculation should be done in order to see.

Thanks

A sentence and a formula were added to specified it in line 110 to 114. The results indicate that the pressure difference between the top and bottom interfaces is less than 0.004 MPa.

Line 93/94: Is pressure at the top outlet different from the normal pressure?

Accept

No, it has been explained above, the pressure difference between the top and bottom interfaces is little, so the top and bottom interfaces of the reactor are under atmospheric pressure.

Line 101/102: why is quartz sand washed? Please comment on that. Why hydrochloric acid? Is it intended to dissolve any present calcite crystals?

Thanks, it has been added as “The quartz sand is soaked in dilute hydrochloric acid for 24 hours in order to remove impurities,” in line 133 and “to ensure that the hydrochloric acid has been removed” in line 136.

Line 115-117: These sentences are hard to understand and restructuring would be helpful.

Thanks, it has been refined in line 149 to 152 as “the quartz sand was dried at 45 °C for 24 hours after washing with pure water.  The white crusted material on the surface of the sand was carefully collected and then subjected to an X-ray diffraction (XRD) analysis in Guangzhou Institute of Energy Conversion (GIEC), CAS.”

Line 166-168: This section may be divided by subheadings. It should provide a concise and precise description 166 of the experimental results, their interpretation as well as the experimental conclusions that can be 167 drawn.” This is leftover from the template I believe.

Accept, it has been deleted.

Line 173: This graph still has Chinese labels instead of English

This figure is very hard to understand. Which is the experimental data and what are the modelled results? Please change this figure accordingly.

Sorry for that, it has been modified.

Line 204/205: Which database was used for PHREEQC? Was the system treated as at equilibrium with the atmosphere? Also, reference for PHREEQC is missing.

As mentioned above, the description of PHREEQC is added and We do not consider the system need treated as at equilibrium with the atmosphere since the contact time between the solution and the air during the experiment is very short.

Line 243: in the brackets, the reference to Luff should be changed into the normal reference format

Sorry for that, it has been modified.

214/242: Two times referencing Luff et al in the same sentence

Author contributions needs to be revised, there seem to be typos there.

Thanks, it has been modified.

Reviewer 2 Report

Review for ijerph-46960

This study is a combination of an experimental and modelling study in order to investigate the formation of authigenic carbonate minerals in combination with methane fluxes. Experimental results were used to estimate the formation of carbonate minerals under cold seep conditions were estimated. This process was then modelled using THROUGHREACT. The approach of combining experiments and modelling is nice, however this manuscript lacks severe description of methods and results.

Comment 1: setup of the fluid cell: it should be commented on how the flow rate affects the observed experimental results. Furthermore, please specify what material the reactor is made out of. Why is there no data of the samples after the precipitation happened? No XRD data or results are presented here even though it was stated that XRD was used. Also, it is mentioned that 8 samples were run, however no comment regarding the variability of the experimental results is given. Furthermore, it needs to be commented on how the solutions were prepared (what is the purity grade of the chemicals used? Was the NaHCO3 stock solution equilibrated with the atmosphere prior to the experiment?) Both impurities and exchange with the atmosphere can have severe effects on solution composition and hence the experimental results.

Comment 2: the saturation index with respect to calcite can be calculated which then can be used in the rate equation. It would be useful for the geochemical reader to specify what the saturation index in these experiments is

Detailed comments:

Line 15-16 “According to the experimental results, the formation process of carbonate minerals under cold seep conditions was estimated. That 1m carbonate need 12000 and 7000 respectively under fast and slow emission conditions”

These two sentences could be combined.

Line 20: unit is missing, carbonate need 17000 and 9700 years I assume to grow

Line 47/48: There is a variety on mineral growth studies on calcite that should be at least mentioned here, some of them are in porous media

e.g. Godinho and Williams, 2018, Geochimica & Cosmochimica https://doi.org/10.1016/j.gca.2017.10.024

Bracco, J.N., Stack, A.G. and Steefel, C.I., 2013. Upscaling calcite growth rates from the mesoscale to the macroscale. Environmental science & technology, 47(13), pp.7555-7562.

Zuddas, P. and Mucci, A., 1998. Kinetics of calcite precipitation from seawater: II. The influence of the ionic strength. Geochimica et Cosmochimica Acta, 62(5), pp.757-766.

Zuddas, P. and Mucci, A., 1994. Kinetics of calcite precipitation from seawater: I. A classical chemical kinetics description for strong electrolyte solutions. Geochimica et Cosmochimica Acta, 58(20), pp.4353-4362.

There are also other studies on kinetics of calcite precipitation and cold seeps:

Karaca D, Hensen C, Wallmann K. Controls on authigenic carbonate precipitation at cold seeps along the convergent margin off Costa Rica. Geochemistry, Geophysics, Geosystems. 2010 Aug;11(8).

Luff R, Wallmann K, Aloisi G. Numerical modeling of carbonate crust formation at cold vent sites: significance for fluid and methane budgets and chemosynthetic biological communities. Earth and Planetary Science Letters. 2004 Apr 30;221(1-4):337-53.

It seems to me as the authors should do a more careful literature research for this manuscript.

Line 54: “Previous studies have been carried out all over the world.” This does not say anything. Please change.

Line 67: Self built experimental device description: How is the temperature controlled in this device? Please comment in the manuscript on this.

Line 92: I assume that normal pressure is ambient condition pressure? Please specify what normal pressure means

Line 93: “and the change in pressure was ignored” Here, it would be good to say that the change in pressure was calculated but found to negligible. I think the calculation should be done in order to see.

Line 93/94: Is pressure at the top outlet different from the normal pressure?

Line 101/102: why is quartz sand washed? Please comment on that. Why hydrochloric acid? Is it intended to dissolve any present calcite crystals?

Line 115-117: These sentences are hard to understand and restructuring would be helpful.

Line 166-168: “This section may be divided by subheadings. It should provide a concise and precise description 166 of the experimental results, their interpretation as well as the experimental conclusions that can be 167 drawn.” This is leftover from the template I believe.

Line 173: This graph still has Chinese labels instead of English

This figure is very hard to understand. Which is the experimental data and what are the modelled results? Please change this figure accordingly.

Line 204/205: Which database was used for PHREEQC? Was the system treated as at equilibrium with the atmosphere? Also, reference for PHREEQC is missing.

Line 243: in the brackets, the reference to Luff should be changed into the normal reference format

214/242: Two times referencing Luff et al in the same sentence

Author contributions needs to be revised, there seem to be typos there.

Author Response

Point1: Idea of laboratory simulation of some processes that happen in unavailable environments (e.g. deep sea) is brilliant. However, the strict conditions that are smilar to natural as much as posible must be taken. Authors wrote: "the formation of cold seep carbonates occurs in the natural high-pressure and low-temperature marine environment, which is a complex biogeochemical process", whereas the conditions of experiment are different. Then, these discrepances are sincerly shown in Conclusions. Paper is not acceptable in this form. There would be good to change the aim of current experiment or add other factors to improve the experiment, to make it more suitable for cold seep environment.

Thanks

This study starts with the relatively simple conditions, such as under normal pressure and without microorganisms involved. However, the experimental data under such conditions could shed some light on the carbonate participation patterns under different flow rates. The main focus of this paper is the effect of flow rate and ion concentration on the precipitation of cold seep calcite. In the future experiments, the combined effects of microbes and pressure would discussed. This study can also provide a comparison for experiments with microbial participation in the future to explore the role of microorganisms in the experimental process. Some similar experiments were also performed under normal pressure, such as [1], and he sampling error may increase under high pressure conditions.

We are in the exploratory phase of the experimental approach and plan to conduct experiments involving microbial and high pressure conditions in the future.

1.    Steeb, P., P. Linke, and T. Treude, A sediment flow-through system to study the impact of shifting fluid and methane flow regimes on the efficiency of the benthic methane filter. Limnology and Oceanography: Methods, 2014. 12(1): p. 25-45.

Round 2

Reviewer 1 Report

Experimental studies and numerical simulations are often only  way of recognition of deep sea conditions. For that reason, the paper is novel and original. It will be interesting especially for specialists.